

# WRF4PALM v1.0: A Mesoscale Dynamical Driver for the Microscale PALM Model System 6.0

Dongqi Lin[1,2], Basit Khan[3], Marwan Katurji[2], Leroy Bird[4], Ricardo Faria[5], and Laura E. Revell[1]

[1]School of Physical and Chemical Sciences, University of Canterbury, Christchurch, New Zealand
[2]School of Earth and Environment, University of Canterbury, Christchurch, New Zealand
[3]Institute of Meteorology and Climate Research, Atmospheric Environmental Research (IMK-IFU), Karlsruhe Institute of Technology (KIT), Garmisch-Partenkirchen, 82467, Germany
[4]Bodeker Scientific, Alexandra, New Zealand
[5]Oceanic Observatory of Madeira, Agência Regional para o Desenvolvimento da Investigação Tecnologia e Investigação, Madeira, Portugal

**Correspondence:** Dongqi Lin (dongqi.lin@pg.canterbury.ac.nz)

**Abstract.** A set of Python-based tools, WRF4PALM, has been developed for offline-nesting of the PALM model system 6.0 into the Weather Research and Forecasting (WRF) modelling system. Time-dependent boundary conditions of the atmosphere are critical for accurate representation of microscale meteorological dynamics in high resolution real-data simulations. WRF4PALM generates initial and boundary conditions from WRF outputs to provide time-varying meteorological forcing
for PALM. The WRF model has been used across the atmospheric science community for a broad range of multidisciplinary applications. The PALM model system 6.0 is a turbulence-resolving large-eddy simulation model with an additional Reynolds-averaged Navier–Stokes (RANS) mode for atmospheric and oceanic boundary layer studies at microscale (Maronga et al., 2020). Currently PALM has the capability to ingest output from the regional scale Consortium for Small-scale Modelling (COSMO) atmospheric prediction model. However, COSMO is not an open source model which requires a licence agree-
ment for operational use or academic research (http://www.cosmo-model.org/). This paper describes and validates the new free and open-source WRF4PALM tools (available on https://github.com/dongqi-DQ/WRF4PALM). Two case studies using WRF4PALM are presented for Christchurch, New Zealand, which demonstrate successful PALM simulations driven by meteorological forcing from WRF outputs. The WRF4PALM tools presented here can potentially be used for micro- and mesoscale studies worldwide, for example in boundary layer studies, air pollution dispersion modelling, wildfire emissions and spread,
urban weather forecasting, and agricultural meteorology.

## 1 Introduction

Over the last decade, research in numerical weather and climate simulations, environmental modelling, and agricultural and urban meteorology, has developed to include higher spatial resolutions, such that the feedback from the microscale (from $10^{-2}$ to $10^3$ m; from seconds to hours) processes impacted by surface heterogeneities can be explicitly resolved and better
represented. At the mesoscale (from $10^4$ to $5 \times 10^5$ m; from hours to days), numerical weather prediction (NWP) models are widely used to simulate regional atmospheric flows in real meteorological conditions. Mesoscale NWP models are primarily





Reynolds Averaged (Navier-Stokes) Simulation (RANS) models that parameterise turbulence without discrepancy for scale (Sagaut, 2006, chapter 1.4). The parameterisations applied in RANS models only consider the average properties of atmospheric flows impacted by surface geometries at the grid resolution of the simulation. In contrast to RANS models, Large Eddy

Simulation (LES) models apply a local spatial filter to solve 3-D prognostic equations (Sagaut, 2006, chapter 1.4). Eddies with scales smaller than the filter (sub-grid scales) are parameterised while eddies larger than the filter are termed as large eddies and are resolved explicitly. LES models have been used to simulate and understand airflows around urban canopy structures at scales of several meters (hereafer fine-scale). For example, Bergot et al. (2015) applied the LES technique embedded in the non-hydrostatic anelastic research model Meso-NH to study fog life cycle and dispersion stability at fine-scale, Wyszogrodzki

et al. (2012) used LES-EULAG to simulate fine-scale urban dispersion, and Kurppa et al. (2020) used the Parallelised Large Eddy Simulation Model (PALM) to analyse spatial distributions of aerosols.

Although LES models are known to have better performance than RANS models when addressing transport and dispersion problems (Gousseau et al., 2011), mesoscale flows still have significant impact on the local LES scale. For simulations to represent realistic meteorology with high fidelity it is essential that the effects of mesoscale flows are captured. Therefore, time

varying initial and boundary conditions of the atmosphere are important to achieve realistic atmospheric simulations in LES domains. With a turbulence and building resolving LES model at its core, the PALM model system 6.0 has been used to study atmospheric and oceanic boundary layers for over 20 years (Maronga et al., 2015, 2020). In recent years the PALM model has been extended by implementing PALM-4U (PALM for urban applications) components for application of PALM model in the urban environments. (Maronga et al., 2015, 2020; Heldens et al., 2020). High-resolution (fine-scale) PALM simulations

have proven to be useful for city planners to determine the optimal layout of surface structures, such as buildings, vegetation and pavements, to mitigate adverse air-quality impacts (e.g. Gronemeier et al., 2017; Kurppa et al., 2018, 2020). However, the studies by Gronemeier et al. (2017) and Kurppa et al. (2018), only performed idealised simulations where the direction and intensity of the wind at inflow were invariant during the entire simulation period. In addition, their simulation domains have to be reoriented to accommodate the impact of wind direction, which can lead to a large amount of additional manual data

processing.

PALM was designed to seamlessly apply forcing from other mesoscale models in an offline-nesting approach (Maronga et al., 2015; Vollmer et al., 2015; Heinze et al., 2017; Maronga et al., 2020). Here offline-nesting is realised as that mesoscale dynamical data are passed onto PALM after mesoscale simulations are finished, while PALM does not have to run along with or provide any feedback to the mesoscale model. Currently, the PALM model system 6.0 provides the additional software package

INIFOR (Maronga et al., 2020) which can process mesoscale data for use by PALM. However, INIFOR is currently configured to only process data output from the regional weather prediction model COSMO (Consortium for Small-scale Modelling), formerly named as LM-K (Baldauf et al., 2007). Vollmer et al. (2015) successfully used COSMO and PALM to reproduce an offshore wind turbine wake in Germany. However, at present, the COSMO model is not an open-source model which can therefore not be directly applied to most regions outside of the European domain. Kurppa et al. (2020) used mesoscale data from

Meteorological Cooperation on Operational Numerical Weather Prediction (MetCoOp) Ensemble Prediction System (MEPS,


Bengtsson et al., 2017; Müller et al., 2017), operated by the Norwegian Meteorological Institute, to provide realistic boundary conditions in PALM. Similar to COSMO, MEPS is currently not publicly available.

To extend the use of PALM for the scientific community, we have developed a set of Python tools to allow PALM to include mesoscale forcing from the Weather Research and Forecasting modelling system (WRF; http://www.wrf-model.org;

Skamarock et al., 2019). These tools are hereafter referred to as WRF4PALM, i.e. tools that process WRF output for use in PALM simulations. The free open-source WRF model has been extensively used for atmospheric research and weather forecasting throughout the world (Skamarock et al., 2019). Using WRF4PALM, modellers can offline-nest the PALM model within the WRF model to generate simulations that resolve microscale meteorological dynamics.

This paper describes WRF4PALM and presents validation of the tools. The PALM dynamical input data standard is de-

scribed in Sect. 2. A description of the WRF4PALM framework is described in Sect. 3. Sect. 4 shows the validation and initial application of WRF4PALM. Sect. 5 presents conclusions and an outlook for WRF4PALM.

## 2 PALM Offline-Nesting and Dynamical Input

The offline-nesting module embedded in PALM works as an interface between a mesoscale atmospheric model and PALM (Maronga et al., 2020). This interface requires users to provide PALM with a netCDF dynamical driver file as an input (hereafter

referred to as the dynamic driver; to be consistent with PALM documentation), which contains the meteorological forcing and initial profiles of atmospheric state variables extracted from the mesoscale model. The dynamic driver created by WRF4PALM focuses solely on correctly and appropriately interpolating the dynamical fields extracted from WRF to fulfil the input data requirements of PALM.

Following the PALM Input Data Standard (PIDS) (https://palm.muk.uni-hannover.de/trac/wiki/doc/app/iofiles/pids), the dy-

namic driver must include initial vertical profiles of the atmosphere and soil, the lateral and top boundary conditions of the atmosphere and time series of surface pressure (Table 1). Note that the variables listed in Table 1 are based on PIDS v1.9. While some variable names may be changed in future updates of PALM, these can be modified in WRF4PALM code in such cases.

## 3 Methodology

### 3.1 WRF4PALM Framework

The new WRF4PALM (available on https://github.com/dongqi-DQ/WRF4PALM) is based on WRF2PALM initially developed by Faria (2019). Modifications and changes made to WRF2PALM to create WRF4PALM are described in Appendix A.

The data passed from WRF to PALM include velocity fields, thermodynamic components (pressure, temperature, potential temperature and water vapour mixing ratio), soil features, vertical grid structure (geopotential) and geographical information

(Table 2). The code structure of WRF4PALM is shown in Figure 1. WRF4PALM is written in the Python3 programming language. Two major Python scripts comprise WRF4PALM. One is `create_cfg.py` which reads user input and specifies the





PALM domain within the WRF domain using latitude and longitude bounds. The other is `create_dynamic.py` which processes WRF dynamical fields to create the PALM dynamic driver. Detailed step-by-step instructions for running WRF4PALM are given in Appendix B.

The PALM grid configuration prescribes how the WRF output needs to be interpolated onto the PALM grid cells along west-east (`nx`), south-north (`ny`) and bottom-top (`nz`) coordinates and the corresponding grid spacing along each direction (`dx`, `dy` and `dz` respectively). The latitude and longitude of the centre of the PALM domain must be provided to specify the PALM domain location in the WRF domain. By obtaining the aforementioned domain configuration information from users, the `create_cfg.py` script then generates a configuration file containing latitudes and longitudes for the north, south,

east, and west lateral boundaries and a grid configuration for the PALM domain. The configuration file then acts as an input for `create_dynamic.py` to finish the interpolation. WRF4PALM also allows users to apply stretched grid spacing along the z-direction. Identical to parameters used in PALM input parameter list, users must define `dz_stretch_level`, `dz_stretch_factor`, and `dz_max` for vertically stretched grid spacing.

The `create_dynamic.py` script requires users to provide their own WRF output. WRF offers abundance of choices of

parameterisations for microphysics, radiation, surface layer etc. Users also have a high degree of freedom to choose the meteorological data for initialisation, geospatial data and projection of the simulation domain. Although optimal WRF configurations will depend on the user's own research interests, any WRF output containing data described in Table 2 is considered applicable for WRF4PALM.

PALM also requires the start and end time stamps (in YYYY, MM, DD, HH format) and lateral and top boundary conditions

update frequency to be provided by users. The lateral and top boundary conditions can update from every 1 minute to every 6 hours (or more) depending on the temporal frequency of WRF output and the user's own research needs. The thickness of the individual soil layers (`dz_soil`) to be used in PALM must be specified in `create_dynamic.py`. The default eight-layer configuration of WRF4PALM is the same as that described in Maronga et al. (2015).

Because both PALM and WRF use the Arakawa Cartesian grid staggering (staggered Arakawa C-grid, Harlow and Welch,

1965; Arakawa and Lamb, 1977), no transformation is required in PALM for staggered data. After reading the input parameters described above, the script first extracts WRF data for the specified period and location required for PALM. Potential temperature, air temperature and pressure fields are read using the `getvar` function embedded in the WRF-Python package (Ladwig, 2019). Other variables, such as water vapour mixing ratio and wind field, are read using the xarray package (Hoyer and Hamman, 2017). Other than the vertical component of wind ($w$), all WRF variables are first interpolated on each horizon-

tal field from the WRF domain onto the PALM horizontal Cartesian grid. The horizontal interpolation uses the SciPy package (Virtanen et al., 2020). The WRF data that were horizontally interpolated to the PALM grid are then vertically interpolated onto PALM vertical Cartesian physical height levels. This requires the `interplevel` function in the WRF-Python package (Ladwig, 2019), which reads the WRF physical height levels and interpolates the given data onto required PALM vertical levels as defined in the PALM domain configuration file created by `create_cfg.py`. The WRF physical height levels are

calculated using:

$$z = (PH + PHB)/g, \tag{1}$$





where PH is the perturbation geopotential in $\mathrm{m^2\ s^{-2}}$, PHB is the base-state geopotential and $g$ is gravitational acceleration ($9.81\ \mathrm{m^2\ s^{-2}}$). Equation (1) only gives staggered vertical height levels which are then destaggered using the `destagger` function in the WRF-Python package (Ladwig, 2019) for vertical interpolation of variables which are not vertically staggered.

When WRF is operated under RANS mode, the value of the vertical component of wind ($w$) does not represent any turbulence and could be very small in WRF. These small values may lead to possible missing values in the horizontal and subsequently the vertical interpolation. In order to avoid this issue, the vertical component of wind ($w$) from WRF is first interpolated vertically to PALM vertical-staggered Cartesian physical height levels using staggered vertical height levels calculated by Equation (1) and `interplevel` function in the WRF-Python package (Ladwig, 2019). Then, the vertically interpolated data

are interpolated horizontally to PALM horizontal Cartesian grid. Once all the interpolation processes of the aforementioned variables are completed, geostrophic winds are calculated assuming geostrophic balance:

$$v_g = \frac{1}{\rho f}\frac{\Delta P}{\partial x}, \tag{2}$$

$$u_g = -\frac{1}{\rho f}\frac{\Delta P}{\partial y}, \tag{3}$$

where $\rho$ is air density in $\mathrm{kg\ m^{-3}}$, $f$ is the Coriolis parameter, $P$ is pressure, $x$ and $y$ are coordinates along west-east and

south-north respectively.

The height levels in WRF are terrain-following (Skamarock et al., 2019) while the Cartesian topography in PALM allows for explicitly resolving obstacles such as buildings and orography (Maronga et al., 2015). Due to such difference in topography representation, the vertical interpolation can lead to Not a Number (NaN) values near the ground surface when WRF data are vertically interpolated onto PALM vertical levels below the first WRF model level. It would be inefficient to create NaN masks

and filter data to fit the entire topography and all surface geometries in PALM simulations. Hence, the surface NaN solver is applied to fill the NaN values near the surface. For all the scalar variables and the vertical velocity ($w$), the surface NaN values are filled by taking the values from the lowest level where valid values exist at the grid point. For horizontal components of velocity ($u$ and $v$), a logarithmic fit is applied. After solving surface NaN values, the initial vertical profiles are calculated by taking the horizontal average of velocity components, potential temperature and water vapour mixing ratio at the initial

time for each vertical level. The time series of surface pressure in the dynamic driver is the time series of horizontal average of pressure at the lowest level after interpolation. Soil moisture and temperature are interpolated to soil layers provided by users. Due to the difference in the grid resolution and data sources between WRF and PALM, all the soil moisture for water bodies in WRF (where soil moisture is equal to 100%) is replaced by the median value of land soil moisture in the given PALM simulation domain to avoid mismatch between the PALM and WRF landmasks. To take into account water bodies in

PALM, a netCDF static driver input file is required (Heldens et al., 2020), where the types and locations of water bodies are specified. Details about static files used in this work are presented in Sect. 3.2. WRF4PALM also gives the soil information at each soil layer (`soil_moisture`, `soil_temperature` and `deep_soil_temperature`; identical to PALM input parameters), which can be added into the input parameter list of the PALM land surface model. Details of algorithms applied in WRF4PALM are provided in the online documentation of WRF4PALM.





The dynamic driver of PALM generated by the `create_dynamic.py` script only contains mesoscale dynamics from WRF and does not encompass any turbulence which is completely parameterized in WRF. In order to obtain realistic flow characteristics, non-cyclic boundary conditions must be applied. When non-cyclic boundary conditions are used with offline-nesting, because no turbulence is included in the inflow, either a large domain is required to allow sufficient space and time for turbulence to develop or the synthetic turbulence generator (STG) must be applied (Gronemeier et al., 2015).

## 3.2   PALM Static Driver

To resolve and realise near surface microscale structures, PALM can read a netCDF static driver file (hereafter static driver) as an input. The static driver includes information on buildings, streets, vegetation, soil, water bodies etc., in the model domain (Heldens et al., 2020). Due to high variability in geospatial data availability and quality across the world, it is unlikely to have a standard automatic process to generate PALM static driver. WRF4PALM does not require users to provide a static driver, which is applicable to both realistic simulations (with static driver) and relatively idealised simulations (without static driver). However, we recommend users to include static driver in PALM simulations for more realistic and representative results to understand the impact of microscale surface structures. In the case studies described later, a static driver is included. We adopted similar procedure described in Heldens et al. (2020) to create the static driver. Various geospatial data sources are used including:

– Digital Surface Model (DSM) and Digital Elevation Model (DEM) from Envirionment Canterbury Regional Council (2020)

– Street and pavement type information from OpenStreetMap (https://planet.openstreetmap.org/)

– Building information from New Zealand building outlines dataset (Land Information New Zealand, 2020a)

– Land cover classification data from Land Cover Database (LCDB) version 5.0, Mainland New Zealand (Landcare Research, 2020)

– Water bodies information from New Zealand parcels dataset (Land Information New Zealand, 2020b)

## 4   Case Studies

Two case studies are presented here to demonstrate the performance, stability and applicability of WRF4PALM. The local solar time, i.e. New Zealand Daylight Time (NZDT), is used for both case studies presented below. Both the case studies were centred on Christchurch airport, New Zealand (43.4864° S, 172.5369° E). Christchurch sits in the Canterbury Plains and in the zone of mid-latitude westerlies in the Southern Hemisphere. A succession of subtropical anticyclones and depressions progress eastwards over the city (Macara, 2016). Westerly flows over Christchurch usually bring high clouds and sunshine while easterly flows sometimes lead to cloudiness and rainfall in Christchurch. The two case studies shown illustrate two simulation scenarios from real weather events that occurred in Christchurch – representing synoptic forcing from north-westerly airflow and the other


is from easterly and north-easterly flow modulated by a diurnal forcing. In this study, ground-based measurements are obtained from an automatic weather station (AWS) operated by the New Zealand Meteorological Service (MetService) at Christchurch Airport.

## 4.1 Model Configuration

The Advanced Research WRF (ARW) system version 4.0 (Skamarock et al., 2019), WRF4PALM, and PALM model system
6.0 (revision 4550) (Maronga et al., 2015, 2020) are used for the case studies. Figure 2 shows the WRF domain configuration with four domains having horizontal resolutions of 27 km, 9 km, 3 km and 1 km, respectively. WRF is initialised with ERA5 data, the fifth generation of European Centre for Medium-Range Weather Forecasts (ECMWF) atmospheric reanalysis of the global climate (Hersbach et al., 2019). The WRF simulation uses the contiguous US (CONUS) physics suite (Liu et al., 2017). This physics suite includes the microphysics parameterisation developed by Thompson et al. (2008); the Tiedtke cumulus
parameterisation (Tiedtke, 1989; Zhang et al., 2011) (domains D01 and D02 only; see Figure 2); no cumulus parameterisation scheme is applied for domains D03 and D04 (see Figure 2); the RRTMG models (Iacono et al., 2008) for longwave and shortwave radiation; the quasi-normal scale elimination (QNSE) planetary boundary layer (PBL) physics parameterisation (Sukoriansky et al., 2005); the eta similarity scheme (Monin and Obukhov, 1954; Janjić, 1994, 1996, 2001) for surface layer parameterisation and the unified Noah land surface model (Tewari et al., 2016). Both WRF simulations presented in this study
have a spin-up time of at least 24 hours and do not have data assimilation technique applied.

Both of the PALM simulations described in the case studies use the following sub-models embedded in PALM: radiation model, land surface model, urban surface model, plant canopy model, synthetic turbulence generator and offline-nesting module. Since this work only aims to demonstrate WRF4PALM's applicability and give an overview of WRF4PALM's performance, the 1 km grid resolution WRF output is directly processed to 10 m grid resolution PALM. Both of the simulations
use non-cyclic boundary conditions to represent realistic meteorological conditions. STG is used in this study to reduce the adjustment zone size near the boundary and to accelerate turbulence development. When PALM reads synoptic conditions from the dynamic driver and generates turbulence with STG, a small flow adjustment zone near lateral boundaries may still appear in the simulation. Hence, although the domain configurations of the two case studies are identical (details see Table 3), the domain locations are different in order to avoid possible boundary artefacts introduced by STG when comparing model data
with observational data at the AWS site. In both PALM simulations, the lateral and top boundary conditions interpolated from WRF are updated hourly. Rayleigh damping at a factor of 0.01 is used near the top boundary (2900 m) to prevent reflection of gravity waves. The STG is called every second with an adjustment interval of 30 seconds.

## 4.2 North-Westerly Case

During the late afternoon of 13 February 2017, the AWS operated by NZ MetService situated at the Christchurch airport
measured strong north-westerly flows. The PALM simulation domain and the AWS location are shown in Figure 3. The 3.6 km (east-west) × 3.6 km (south-north) simulation domain is designed to: 1) include the AWS in order to compare the model data with the observational data and 2) avoid possible artefacts produced by STG near the north and west lateral boundaries. A





narrow zone of laminar flows near the lateral boundaries at inflow can appear in the simulation due to the flow adjustment zone created by STG. As shown in Figure 4, this PALM simulation includes the 24-hour period between 1500 NZDT 13 February

and 1500 NZDT 14th February 2017, which have seen the sustained north-westerlies over Christchurch. In Figure 4, time series of the WRF modelling data have an hourly temporal interval while both PALM modelling data and observational data are 1 minute averages. Here both PALM and WRF winds are at 10 m in order to compare with 10 m winds measured by the AWS. Both PALM and WRF modelled temperature data shown in Figure 4 are 2-m data to represent surface air temperatures. The time series of wind direction, wind speed and air temperature show good agreement between the observational data and

the modelled data. WRF overestimated wind speed during the first 2 hours shown in Figure 4, and underestimated wind speed between 0500 NZDT and 0800 NZDT 14 February. In addition, the air temperatures simulated by WRF are approximately 2 °C lower than the observed temperature during the entire 24-hour period. Table 4 compares the modelled surface temperature and wind speed with the observational data. The root-mean-square errors (RMSE):

$$RMSE = \sqrt{\frac{1}{n}\sum_{i=1}^{n}[F_i - O_i]^2}, \tag{4}$$

and index of agreement (IOA)

$$IOA = 1 - \frac{\sum_{i=1}^{n}|F_i - O_i|}{\sum_{i=1}^{n}(|F_i - \bar{O}| + |O_i - \bar{O}|)} \tag{5}$$

are used for the comparison between modelled and observational data, where $F_i$ ($i = 1, 2, \ldots, n$) indicates model estimates or predictions, $O_i$ ($i = 1, 2, \ldots, n$) indicates the pair-wise-matched observations, and $\bar{O}$ is the mean value of observations (for details of the equations see Equation 2 in Chai and Draxler (2014) and Equation 3 in Willmott et al. (2012), respectively). Both

RMSE and IOA are measures of the degree of model prediction error. The smaller the RMSE is, the better the model fits to the observations. In terms of IOA, a value of 1 indicates a perfect match while 0 indicates no agreement at all. The calculation is applied to the surface time series data shown in Figure 4. Hourly averages of both PALM-simulated data and the observational data are taken in order to compare them with the WRF-simulated hourly data. Based on RMSE (2.02 for temperature and 2.70 for wind speed) and IOA (0.72 for temperature and 0.50 for wind speed) given in Table 4, WRF results are satisfactory

compared with other WRF studies. For example, the best RMSE and IOA for wind speed in Indasi et al. (2017) are 2.30 and 0.66 respectively while they also have simulations with RMSE of 4.21 and IOA of 0.43; temperature simulated by WRF in Bhati and Mohan (2018) has RMSE between 3.87 and 7.99 and IOA between 0.58 and 0.81. Overall, PALM has smaller RMSE and higher IOA than WRF meaning that PALM has better performance than WRF regarding surface temperature and wind speed estimation in this case. The comparison between PALM and WRF also shows good agreement (IOA of 0.87 and

0.75 for surface temperature and wind speed, respectively). The improvement in observation-related RMSE and IOA by PALM is due to the inclusion of surface geometries and the LES ability for better resolving near-surface turbulence.

Throughout the 24-hour period, the results in PALM align with those from WRF. The time series of surface winds and temperatures in PALM shown in Figure 4 are similar to those in WRF, while PALM shows higher surface temperatures than





WRF before 0500 NZDT 14 February 2017. To further validate the performance of WRF4PALM, comparisons between WRF,
the WRF4PALM dynamic driver and PALM are carried out. Figure 5 compares the profiles of the u-component of winds
between WRF, the WRF4PALM dynamic driver and PALM. The profiles include 1) the initial vertical profiles and 2) left
(west) boundary conditions (south-north vertical cross section at the left boundary). Profiles of other parameters in the dynamic
drivers are not shown here. The WRF profiles are interpolated by WRF4PALM to the dynamic driver, which is further used
as an input for PALM offline-nesting. As shown in Figure 5, profiles in the dynamic driver are generally identical to profiles
in WRF meaning that WRF data are successfully interpolated and processed by WRF4PALM. Differences between profiles in
PALM and WRF can be spotted (Figure 5), which are due to the turbulence generated by the STG embedded in PALM. Figure 5
shows that WRF4PALM successfully interpolates dynamics from WRF and passes them to PALM through the dynamic driver.
The boundary layer height is automatically calculated in PALM, which is  2300 m in Figure 5d.

The vertical profiles of u-component and v-component of winds and potential temperature ($\theta$) at 1500 NZDT and 2100
NZDT on 13 February 2017 and at 0300 NZDT and 0900 NZDT on 14 February 2017 in PALM are almost identical to
vertical profiles in WRF as shown in Figure 6 and the boundary layer heights in WRF and PALM are consistent over time. The
maximum boundary layer height (MBLH) is around 1.5-1.8 km in both WRF and PALM. The only major difference in the
vertical profiles between the two models can be spotted near the surface. This difference is due to the impact of surface canopy
because PALM is able to explicitly resolve vegetation and building structures in the simulation. A spatial resolution of 10 m may
not be sufficient to represent detailed structures of buildings, but such resolution allows PALM to adequately represent most of
the surface geometries in the model domain. We believe the high similarity between PALM and WRF demonstrates successful
offline-nesting using WRF4PALM. The lateral and top boundaries of PALM are offline-nested with WRF, and the mesoscale
forcings from WRF are updated every hour. Driving an LES model using hourly update cycles from a mesoscale model ensures
mesoscale disturbances are represented, but it may also hinder the microscale boundary layer dynamics developing their own
unique state (Schalkwijk et al., 2015; Heinze et al., 2017). Hence, PALM follows most of the dynamics processed from WRF
and this could be further investigated by relaxing the update cycle of PALM's boundary conditions. In addition to the turbulent
time series of PALM shown in Figure 4, Figure 7 compares the vertical and horizontal cross sections between WRF and PALM.
The WRF cross sections presented in Figure 7 are the nearest four WRF grid points (4 km × 4 km) processed to the PALM
domain (3.6 km × 3.6 km). The cross sections of the two models show that PALM has strong agreement with WRF. However,
WRF is not able to resolve any surface geometries because it is a terrain following model and only simulates mesoscale
characteristics of airflows. On the contrary, PALM's Cartesian grid structure allows PALM to resolve all the terrain structure
and urban canopy explicitly. White patches and areas shown in Figure 7d are buildings and terrains which are higher than 5
m above the lowest level in the PALM simulation domain. Microscale characteristics, such as the local lift and drag forces as
well as turbulence, are only realised in PALM.

To demonstrate and further validate wind anomalies produced in the PALM simulation, Figure 8 compares the modelled
wind speed anomalies in PALM to the anomalies observed by the AWS. The wind anomalies are calculated by differencing
the instantaneous 1-minute wind speed from the hourly averaged wind speed for each hour during the 24-hour simulation
period. As shown in Figure 8, the modelled anomalies vary from approximately -4.2 m s$^{-1}$ to 3.5 m s$^{-1}$ while the observed


anomalies vary from approximately -5.0 m s$^{-1}$ to 4.0 m s$^{-1}$. The standard deviation ($\sigma$) of modelled data (2.662) is greater

than the observational data (2.118). PALM created more positive anomalies but with lower magnitude. The underestimation in the intensity is likely to result from underpredictions in night-time turbulence generated by PALM incorporating the STG at inflow or the biases produced in the model due to coarse grid spacing (van Stratum and Stevens, 2015). The spatial resolution used in the PALM simulation is only 10 m, which may not be sufficient to represent the nocturnal boundary layer properly. Despite the underestimation, PALM is able to reproduce the wind trends and directions. The wind anomalies statistics of WRF

are not shown here because 1) the RANS mode of WRF only presents average properties of airflows and 2) the WRF output used here only contains hourly data, which cannot give any wind anomaly information at each hour during the simulation period.

### 4.3   North-Easterly Case

In the late afternoons on 15 February 2017, easterly-north-easterly flows were observed over Christchurch airport. During the

early mornings on 16 February 2017, calm northerlies were recorded. Similar to the north-westerly case described in Sect. 4.2, the PALM simulation domain (see Figure 9) is designed to include the AWS and avoid artefacts near the north and east lateral boundaries. Figure 10 shows time series of wind direction, wind speed, air temperature and cloud height during the 24-hour PALM simulation period from 1700 NZDT 15 February 2017 to 1700 NZDT 16 February 2017. Similar to the north-westerly case, profiles in WRF, the WRF4PALM dynamic driver and PALM are consistent (Figure 11). The vertical profiles

shown in Figure 12, the vertical and horizontal cross sections shown in Figure 13 all show good agreement between PALM and WRF. The MBLH for this case is 900 m in both WRF and PALM. However, PALM does not predict surface temperatures and winds as well as in the north-westerly case described above. For the period after 0700 NZDT 16 February (see Figure 10), PALM follows the increasing trends of surface temperature and wind speed in WRF, but the underestimation of surface temperature in PALM is significant. Wind speed in PALM is approximately 2 m s$^{-1}$ lower than both the WRF modelled data

and the observational data. The largest difference in surface temperature between PALM and both WRF and the observational data is approximately 7 °C. In terms of the RMSE and IOA for this case shown in Table 4, PALM has worse scores than WRF, despite the fact that PALM still has adequate agreement with WRF (IOA of 0.66 and 0.79 for surface temperature and wind speed, respectively). The wind anomaly analysis for the hourly averaged wind speed during the entire 24-hour simulation period is shown in Figure 14. In this case, PALM only has an adequate performance in terms of modelling wind anomalies.

Similar to the north-westerly case, anomalies simulated by PALM have more positive and smaller values. Due to the bias in the surface wind and temperature, PALM also underestimated wind anomalies significantly. The modelled anomalies vary from approximately -2.8 m s$^{-1}$ to 2.0 m s$^{-1}$, while the observational data show that the anomalies have a range approximately between -3.2 m s$^{-1}$ to 2.8 m s$^{-1}$. The standard deviation also shows underestimation in PALM (2.238) compared with the observational data (2.953).

There could be several reasons for the bias in PALM. Regardless of different initialization situations between the two case studies, cloud cover is suspected to be one particular reason causing errors in PALM. In both PALM simulations, the clear-sky radiation scheme is used. This radiation scheme is the simplest scheme in the PALM modelling system and neglects all clouds.



The observed cloud height and WRF-modelled clouds are shown in Figures 4 and 10. Here the variable cloud fraction is used to represent cloud cover in WRF. The grey shaded periods in Figures 4 and 10 represent when cloud fraction in WRF is greater

than zero. Cloud fractions in WRF were averaged over the closest ten grid cells over the Christchurch airport. In the north-westerly case, most of the simulation period saw clear skies and only a small number of high clouds (above approximately 7500 m) were observed above the Christchurch airport and WRF generally has correct prediction of cloud cover (see Figure 4). In contrast, the period between 0400 NZDT and 1700 NZDT 16 February saw sustained low level clouds (1000 m to 3000 m) (see Figure 10). WRF may have managed to have an adequate estimation of cloud cover during early mornings on 16 February and

hence has a better performance than PALM in this north-easterly case. Because no clouds are simulated in clear-sky PALM, the simulated radiation in PALM becomes unrealistic. According to Table 4, the RMSE and IOA of surface temperature in PALM for clear sky periods (before 0400 NZDT 16 February) are considerably better than the numbers for cloudy periods (after 0400 NZDT 16 February). Another possible reason for PALM's poor performance could be the internal dynamics in PALM. As shown in Figure 10, PALM simulated a north-westerly airflow near the AWS site near 0800 NZDT 16 May 2017.

The north-westerly air mass results in a more convective surface and significant decrease in surface temperature in the west part of the PALM simulation domain (not show). In this north-easterly case, the wind speed during the simulated period is generally low, which cannot offset the convections in PALM domain. We believe the validation results of the PALM simulations with observations are not related to the technique of the offline-nesting or WRF4PALM. Rather, the dynamics, weather conditions, or PALM domain configurations may be the possible factors. The PALM domain location in this case is different from the

north-easterly case and the sensitivity of the domain grid spacing or domain size may also need to be evaluated. Further studies are required to investigate why PALM underestimates surface temperature and wind speed. However, this is beyond the scope of this study as here we only aim to validate WRF4PALM. Although PALM provides several radiation scheme options and has the bulk cloud module embedded, the detailed dynamics in simulations may vary case by case and hence the optimal simulation setup of PALM must be examined in future simulations.

## 340 5 Conclusions

This study describes a utility WRF4PALM that is developed to generate mesoscale forcing from WRF output for the PALM model system 6.0. Results of the application are also validated by two case studies in Christchurch, New Zealand for summer season. WRF4PALM does not require users to pre-process any data manually, but users need to provide their WRF output and PALM domain configuration. WRF4PALM only encompasses mesoscale dynamics from WRF and does not require a

static driver of PALM. In order to include surface heterogeneities, the PALM static driver is used in this study when it is necessary to realise microstructures in urban environment and hence to achieve realistic and representative results in PALM simulations. In the case studies, WRF4PALM was applied to two weather events simulated by WRF for a north-westerly case and a north-easterly case in Christchurch, New Zealand. The case studies are designed to demonstrate the numerical stability of WRF4PALM rather than properly validate meteorology in the simulations. As shown in the case studies, overall WRF4PALM

is considered to have good stability and is able to process dynamics from WRF to PALM successfully. While PALM inherited



most of the characteristics from WRF through the WRF4PALM dynamic driver, PALM's ability to resolve turbulence structure is essential to realise and represent microscale dynamics in urban environment. Comparison of wind anomalies statistics also show satisfactory agreement between PALM and the AWS observational data.

For future use, the domain size, grid spacing, radiation scheme and all other model setup of WRF and PALM need to be evaluated subject to users' own objectives and scope of research questions. In this study, the lateral and top boundary conditions in the WRF-PALM offline-nesting are updated every 1 hour meaning PALM is significantly impacted and constrained by WRF. The sensitivity to the relaxation time and the STG configuration to develop initial turbulence need further assessment.

WRF4PALM is distributed as a free and open-source tool. In the future development, we aim to optimise WRF4PALM in terms of computation time and to further automate the process. As described in the PALM input data standard, the dynamic driver of PALM can also include boundary conditions of chemistry species, such as $PM_{10}$, $NO_x$ ($NO_2$, NO) and $SO_4$ etc., to simulate air pollution in urban environments. WRF4PALM has the potential to process chemistry data from WRF-Chem (the WRF model coupled with chemistry, Grell et al., 2005) to PALM using the similar interpolation technique described in Sect. 3.1. Although several tests and validations have been carried out for WRF4PALM, they may not be conclusive. To improve and extend the use of WRF4PALM, we welcome all users to optimise, modify and contribute to the code.

*Code availability.* The WRF model system V4.0 and the WRF Pre-processing System (WPS) V4.0 used in this study are free and open-source numerical atmospheric modelling systems (https://www.mmm.ucar.edu/weather-research-and-forecasting-model). The PALM model system 6.0 (revision 4550: https://palm.muk.uni-hannover.de/trac/browser?rev=4550) used in this study is free-available online (http://palm-model.org) under the GNU General Public License v3. WRF4PALM code is freely available at https://doi.org/10.5281/zenodo.4017005 or https://github.com/dongqi-DQ/WRF4PALM distributed under GNU General Public License v3.0. Details of Python packages used in WRF4PALM are given on the GitHub repository.

*Sample availability.* The WRF4PALM dynamic driver and the corresponding configuration file are available in the supplements for the north-westerly case study described in Sect. 4.2. All PALM input files to simulate the north-westerly case will also be provided in the supplements.

## Appendix A: WRF4PALM Development

Based on ideas and techniques applied in WRF2PALM, the following changes have been made to develop WRF4PALM:

- Add initial vertical profiles interpolated from WRF to initialise PALM

- Modify geostrophic wind calculation

- Add `create_cfg.py` script to read user input to reduce manual processing

- Improve methods for PALM domain configuration, which is almost identical to PALM input parameters





– Interpolation order is modified. Horizontal interpolation is performed before vertical interpolation.

– Adjust physical heights calculated from WRF for vertical interpolation.

– Adjust horizontal interpolation method for boundary conditions.

– Adjust PALM domain height calculation for vertical interpolation and calculation of top boundary conditions

– Add staggered coordinates for wind field ($u$, $v$ and $w$) interpolation

– Add 3-D soil moisture and soil temperature profiles interpolated from WRF

– Allow PALM domain size smaller than one WRF grid cell size

– Allow users choose simulation period and update frequency based on WRF model outputs

– Add functions to create coordinate information for PALM self-nested domains

– Add surface NaN solver

– Add parameters and functions to enable vertically stretched grid spacing in the dynamic driver and subsequently PALM

RAM (Random Access Memory) usage is modified after the aforementioned development and several other small tweaks made in WRF4PALM. WRF2PALM functions are either removed or modified in WRF4PALM. The following Python functions used in WRF4PALM are based on WRF2PALM:

1. Function to determine the nearest gird cells in WRF to be interpolated to PALM Cartesian grid:

```
def nearest(array, number):
               '''
               find nearest index value and index in array.
               nearest(array, number)
               return(nearest_number, nearest_index)
'''
               import numpy as np
               nearest_index = np.where(np.abs(array-number) ==
                              np.nanmin(np.abs(array-number)))
               nearest_index = int(nearest_index[0])
nearest_number = array[nearest_index]
           return(nearest_number, nearest_index)
```





2. Function to interpolate WRF data horizontally:

```
def interp_array_2d(data, out_x, out_y, method):
    '''
    2d matrix data, x number of points out_x, y number of points out_y,
    method 'linear' or 'nearest'
    '''
    y = np.arange(0, data.shape[0], 1)
    x = np.arange(0, data.shape[1], 1)
    interpolating_function = RegularGridInterpolator((y, x), data,
                                                     method = method)
    yy, xx = np.meshgrid(np.linspace(x[0], x[-1], out_x),
            np.linspace(y[0], y[-1], out_y))
    data_res = interpolating_function((xx, yy))
    return (data_res)
```

3. Function to interpolate 1-dimensional array while calculating geostrophic winds:

```
def interp_array_1d(data, out_x) :
    '''
    1d matrix data, x number of points out_x.
    Output a linear interpolated array
    '''
    x = np.arange(0, data.shape[0], 1)
    xvals = np.linspace(0, data.shape[0], out_x)
    data_res = np.interp(xvals, x, data)
    return (data_res)
```

## Appendix B: WRF4PALM Step-by-Step Guide

A more detailed manual is available on https://github.com/dongqi-DQ/WRF4PALM.

### B1    Step 1 Specify the Domain

Users first need to give the domain information in the `create_cfg.py` script. The information includes:





```
     case_name_d01 = 'chch_10m_NW' # case name as you prefer, but should be
                                   # consistent with the one used in dynamic script
centlat_d01   = -43.487       # latitude of domain centre
     centlon_d01   = 172.537       # longitude of domain centre
     dx_d01        = 10            # resolution in meters along x-axis
     dy_d01        = 10            # resolution in meters along y-axis
     dz_d01        = 10            # resolution in meters along z-axis
nx_d01        = 360           # number of grid points along x-axis
     ny_d01        = 360           # number of grid points along y-axis
     nz_d01        = 120           # number of grid points along z-axis
```

Run `create_cfg.py` to create a configuration file containing the domain information for `create_dynamic.py`.

**B2   Step 2 Process WRF for PALM**

1. Specify case name which should be the same as the one specified in Step 1.

```
     case_name = 'chch_10m_NW' # case name as specified in create_cfg.py
```

2. Specify the WRF output file to process.

```
     wrf_file = 'wrfout_domain_yyyy-mm-dd'
```

3. Specify the start and end time stamp as well as the update frequency of boundary conditions.

```
     dt_start = datetime(2017, 2, 11, 0,) #  start time in YYYY/MM/DD/HH format
     dt_end   = datetime(2017, 2, 12, 0,) #  end time in YYYY/MM/DD/HH format
interval = 2                         #  specify update frequency
```

4. Specify the depth of soil layers.

```
     dz_soil = np.array([0.01, 0.02, 0.04, 0.06, 0.14, 0.26, 0.54, 1.86])
     # this is the default 8-layer setup in PALM
```





5. If stretched vertical grid spacing is desired, specify the following parameters:

```
dz_stretch_factor = 1.02 # stretch factor for a vertically stretched grid
                         # set to 1 if no stretching is desired
dz_stretch_level  = 1200 # Height level above which the grid cells
                         # are to be stretched vertically (in m)
dz_max            = 30   # allowed maximum vertical grid spacing (in m)
```

6. Run `create_dynamic.py` and if successfully executed, a dynamic driver file will be ready.

*Author contributions.* RF provided the WRF2PALM code. BK and DL contributed to the initial development of WRF4PALM. DL contributed to major WRF4PALM development and distribution. LB helped DL with setting up WRF simulation. DL carried out the WRF and PALM simulations and analysed the data. MK and LR supervised DL in performing the case studies. DL wrote the manuscript with contribution from BK, MK and LR. BK, MK and LR reviewed the manuscript.

*Competing interests.* No competing interests are present.

*Acknowledgements.* DL acknowledges support from the University of Canterbury and the Ministry of Business, Innovation and Employment project Particulate Matter Emissions Maps for Cities (Grant no. BSCIF1802). RF was financially supported by the Oceanic Observatory of Madeira Project (M1420-01-0145-FEDER-000001-Observatório Oceânico da Madeira-OOM). We would like to thank Iman Soltanzadeh and Neal Osborne from New Zealand MetService for providing automatic weather observations in Christchurch, New Zealand. We thank Rui Caldeira from Oceanic Observatory of Madeira for internal review of the original manuscript. DL acknowledges Jiawei Zhang at University of Canterbury for his help regarding technical issues related to PALM simulations. All PALM simulations presented in this study were performed on New Zealand eScience Infrastructure (NeSI) high performance computing facilities. WRF simulations were performed on the University of Canterbury (UC) high performance computing cluster.





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





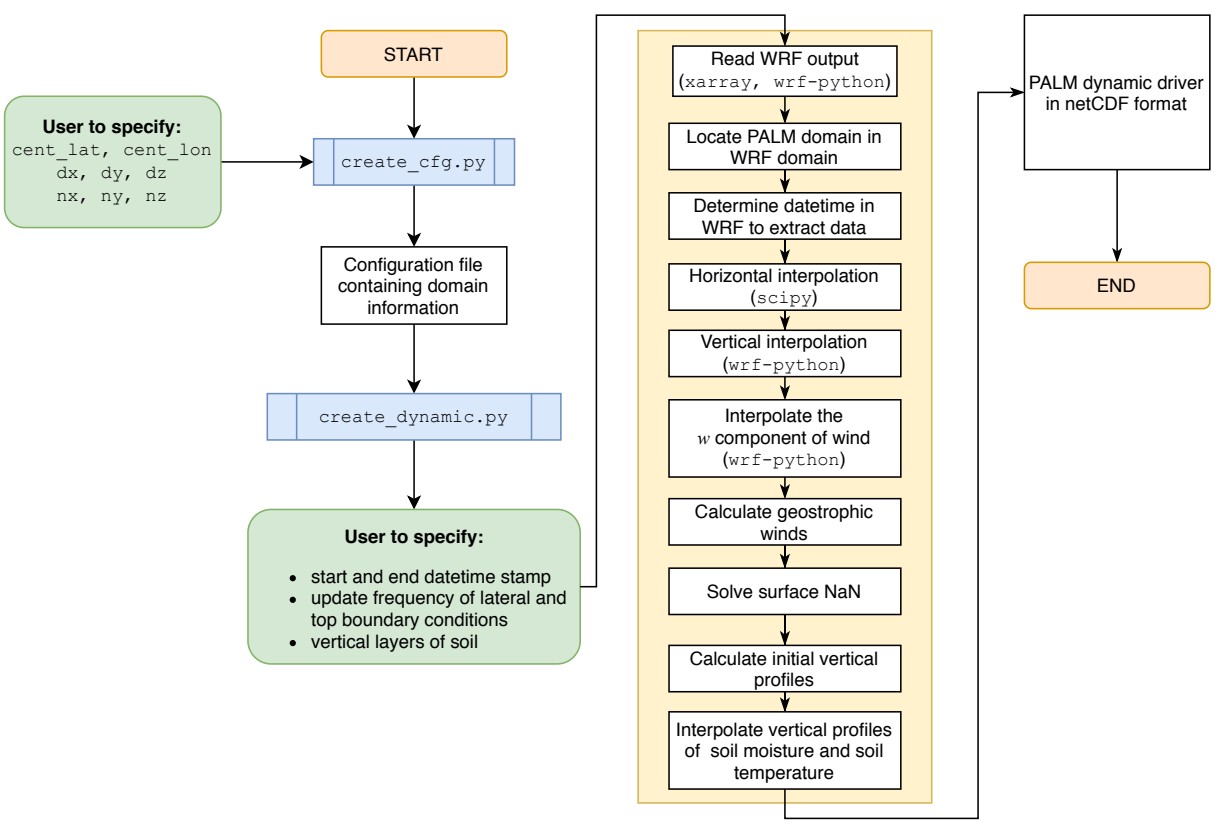

**Figure 1.** The code structure of WRF4PALM. Green boxes indicate input from users. Blue boxes indicate the Python script used. The big yellow box illustrates the processes used to interpolate and pass WRF dynamical fields to the PALM dynamic driver. White boxes outside the yellow box indicate output files of WRF4PALM.

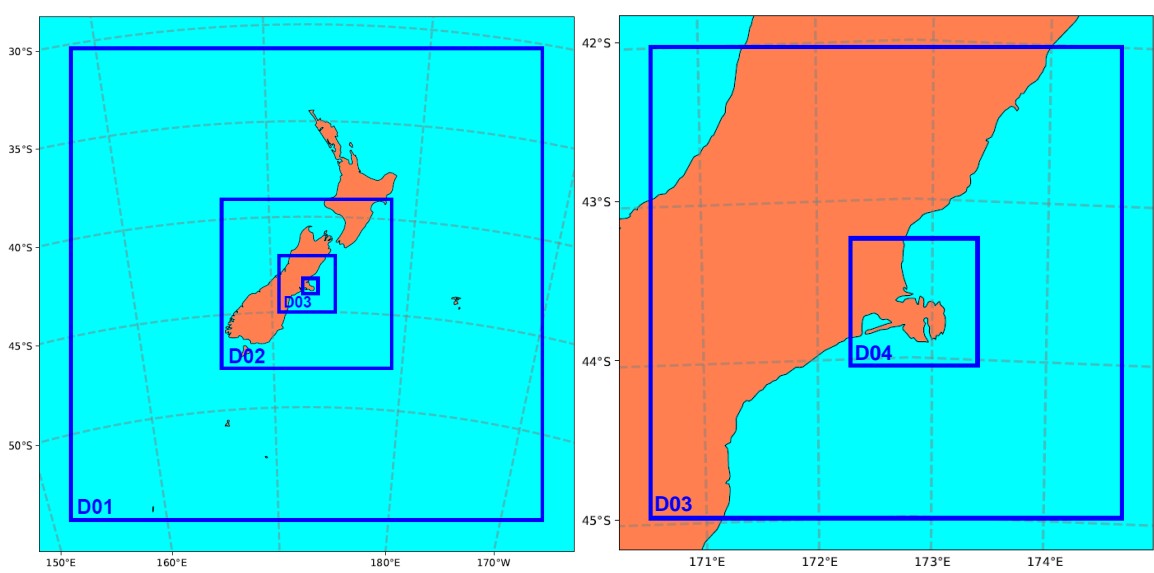

**Figure 2.** WRF domains configuration. Domains D01, D02, D03 and D04 with resolution of 27 km, 9 km, 3 km, and 1 km, respectively (Contains imagery ©Cartopy).



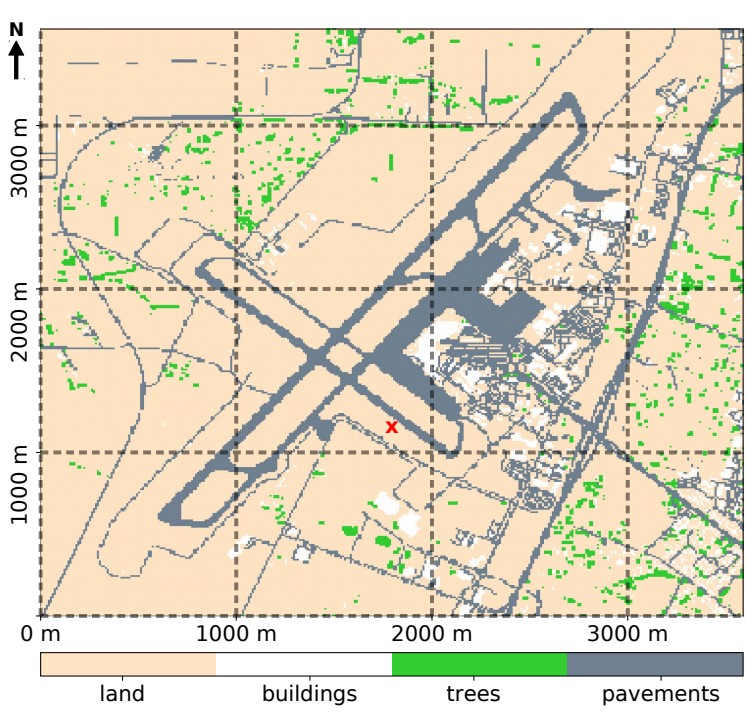

**Figure 3.** PALM modelling domain (3.6 km × 3.6 km) for the north-westerly case. Dotted black lines illustrate the grid size of the WRF model. Buildings are in white, trees are in green, pavements are in grey and other types of the surface are coloured in sand yellow. The red cross shows the AWS location.



**Figure 4.** Time series of simulated (a) wind direction, (b) wind speed and (c) air temperature for the north-westerly case between 1500 NZDT 13th February and 1500 NZDT 14th February 2017 compared with observations. In panel (c) the yellow line indicates cloud height observed at the AWS. In (b) and (c), the shaded grey periods indicate when clouds are simulated in the WRF model.



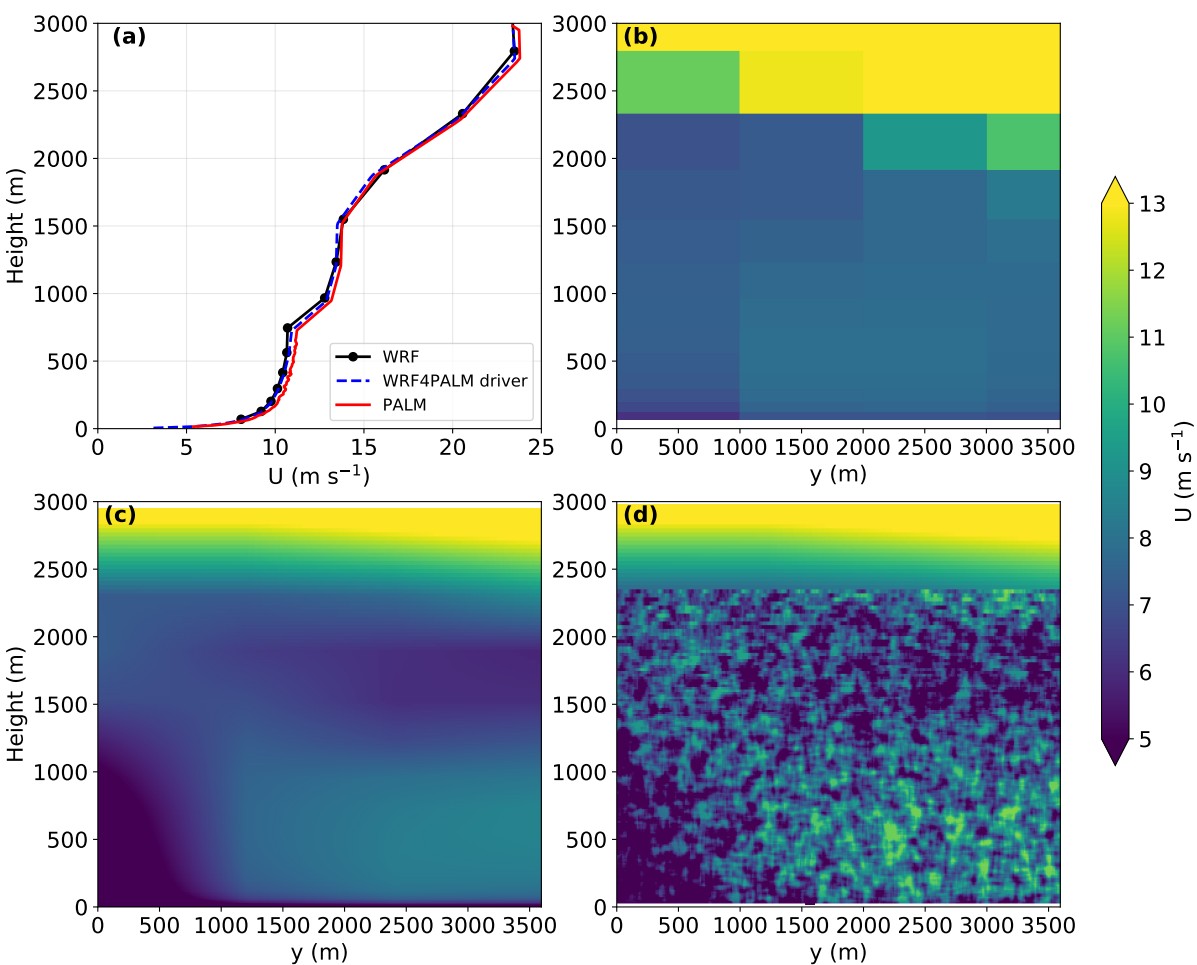

**Figure 5.** (a) Vertical profiles of u-component of winds in WRF, the WRF4PALM dynamic driver and PALM at the initial time. The profiles taken from WRF and PALM are both horizontally averaged. Vertical cross sections of u-component of winds taken from WRF (nearest four grid cells) (b), the dynamic driver (c) and PALM (d) at 1400 NZDT 14 February 2017.



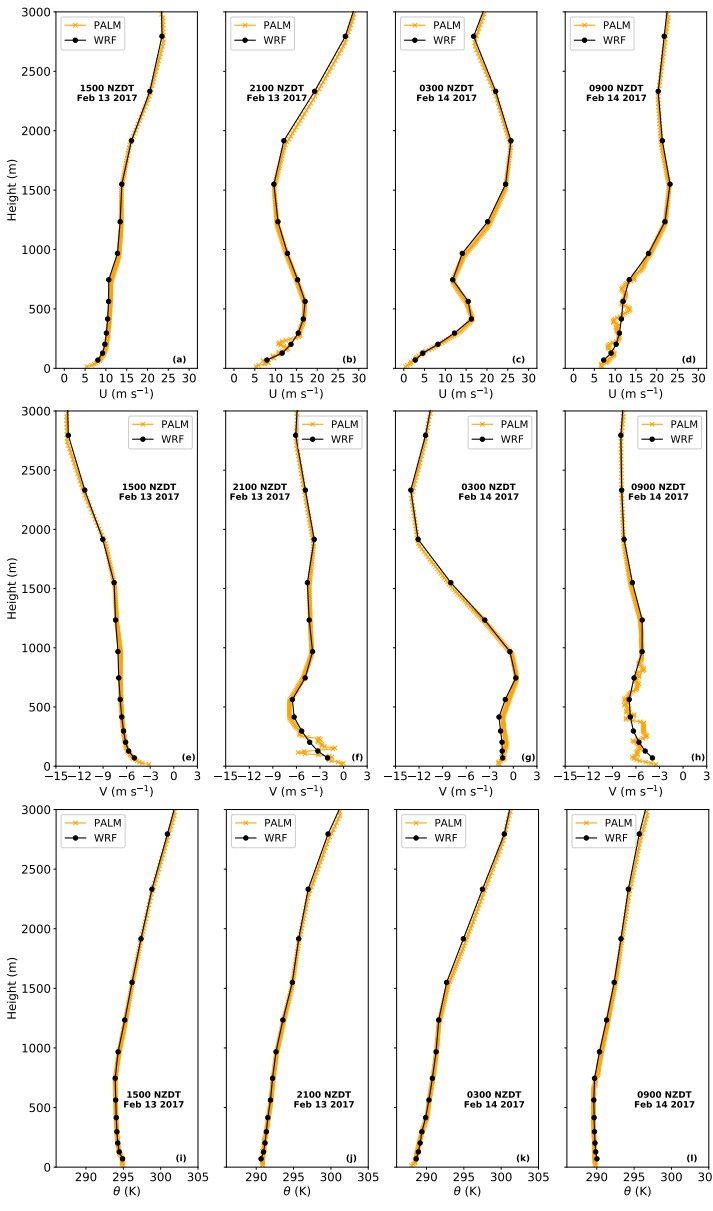

**Figure 6.** The north-westerly case. Vertical profiles of u-component of winds (a, b, c, and d), and v-component of winds (e, f, g, and h), potential temperature ($\theta$) (i, j, k, and l) taken from PALM and the WRF model at the times indicated in the figures (from left to right).



**Figure 7.** The north-westerly case. South-north vertical cross sections of u-component of winds taken from the WRF model (a) and PALM (b) at 1400 NZDT 14 February 2017. Horizontal cross sections of u-component of winds taken from the lowest level in the WRF model (c) and 5 m height in PALM (d) at 1400 NZDT 14 February 2017. White areas indicate terrain and buildings higher than 5 m above the lowest level in the PALM simulation domain.



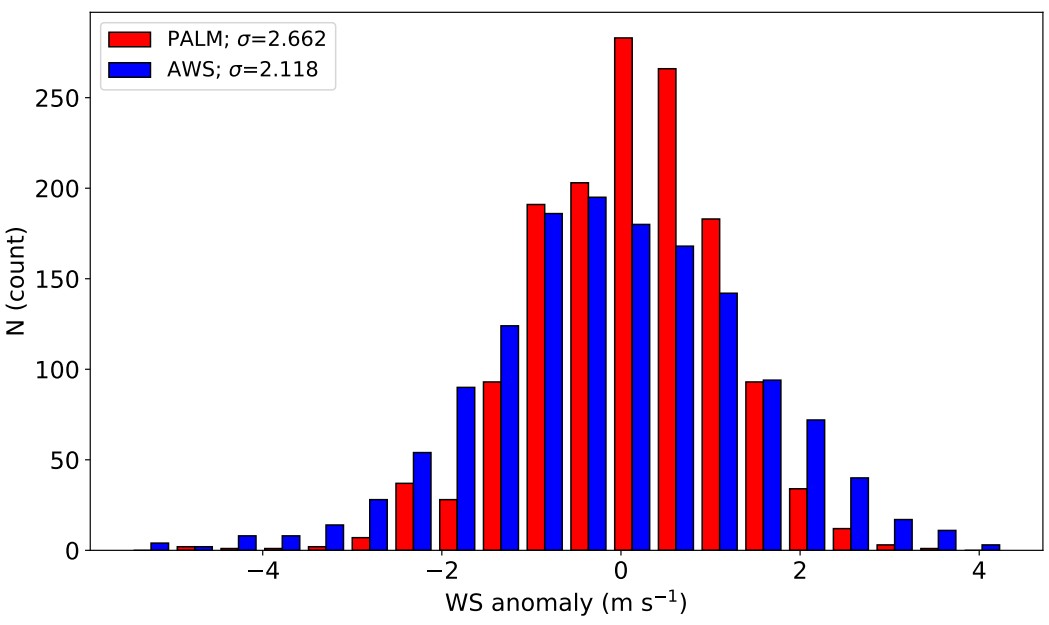

**Figure 8.** Comparison of the range and distributions of hourly wind speed anomalies between PALM and the observational data for the 24-hour period of the north-westerly case. $\sigma$ is the standard deviation of the anomaly data.

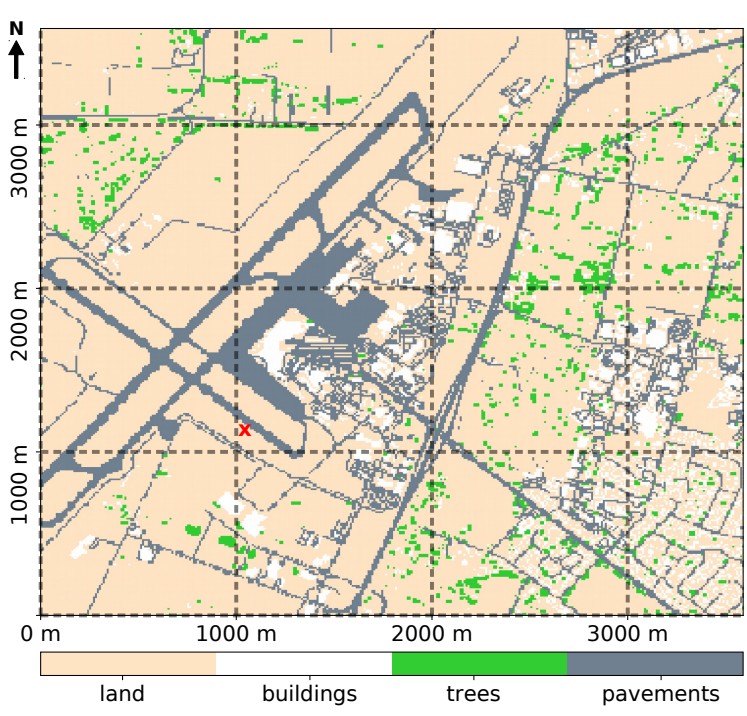

**Figure 9.** As in Figure 3, but for the north-easterly case.







**Figure 10.** As in Figure 4, but for the north-easterly case between 1700 NZDT 15 May 2017 and 1700 NZDT 16 May 2017.



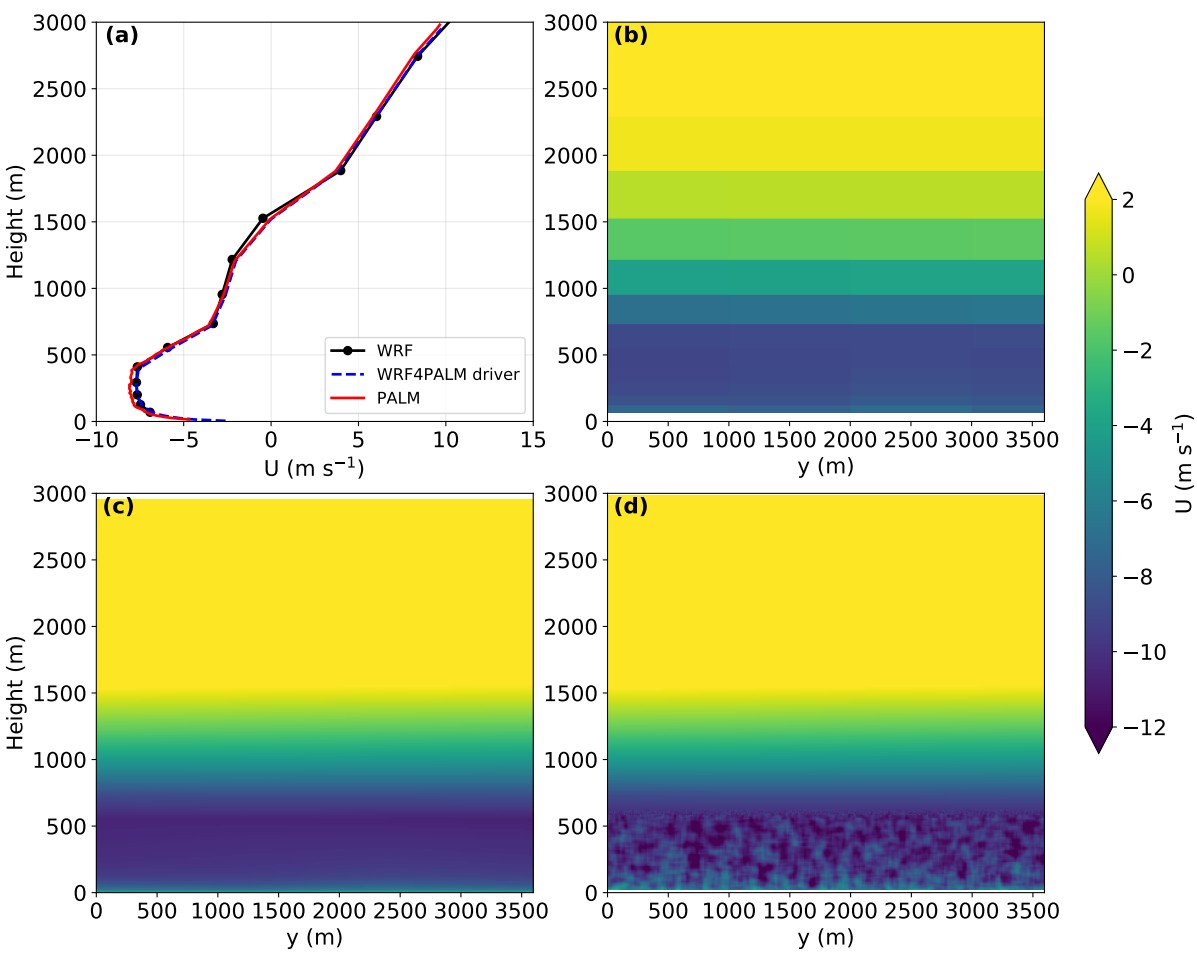

**Figure 11.** As in Figure 5 but for the north-easterly case. (b), (c) and (d) are at 1600 NZDT 16 May 2017.



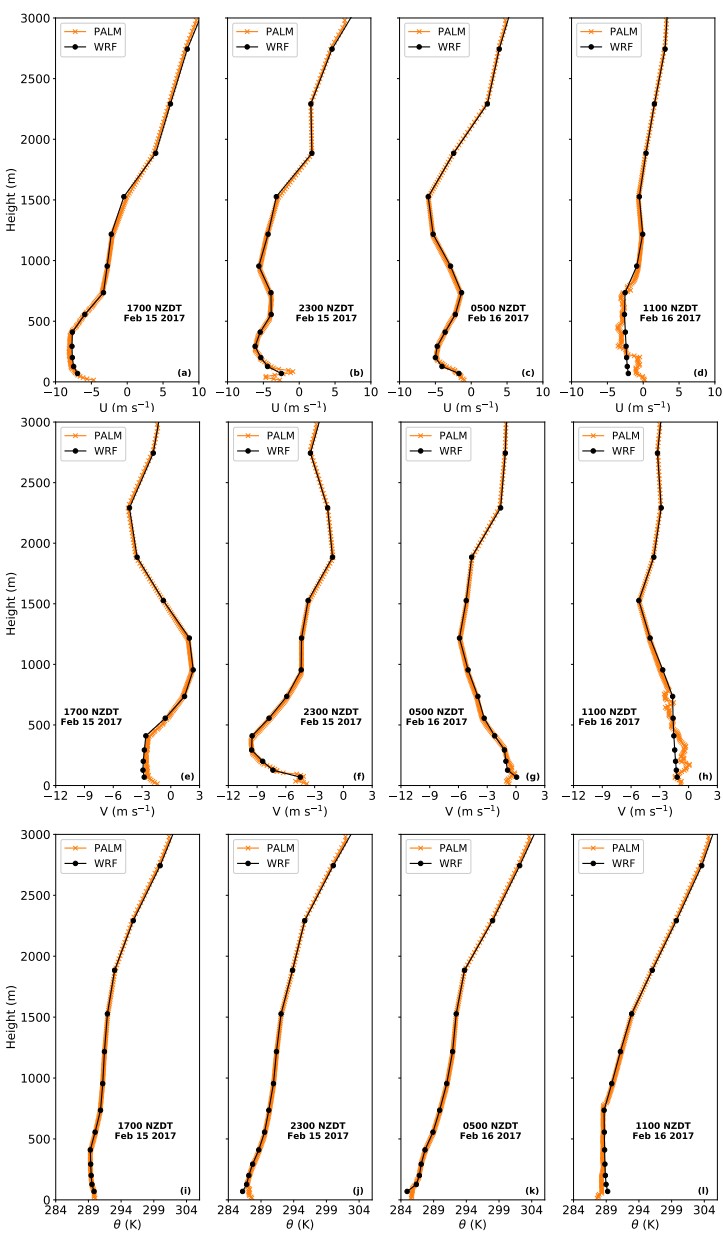

**Figure 12.** As in Figure 6, but for the north-easterly case between 1700 NZDT 15 May 2017 and 1700 NZDT 16 May 2017.



**Figure 13.** As in Figure 7, but at 1600 NZDT 16 May 2017 for the north-easterly case.





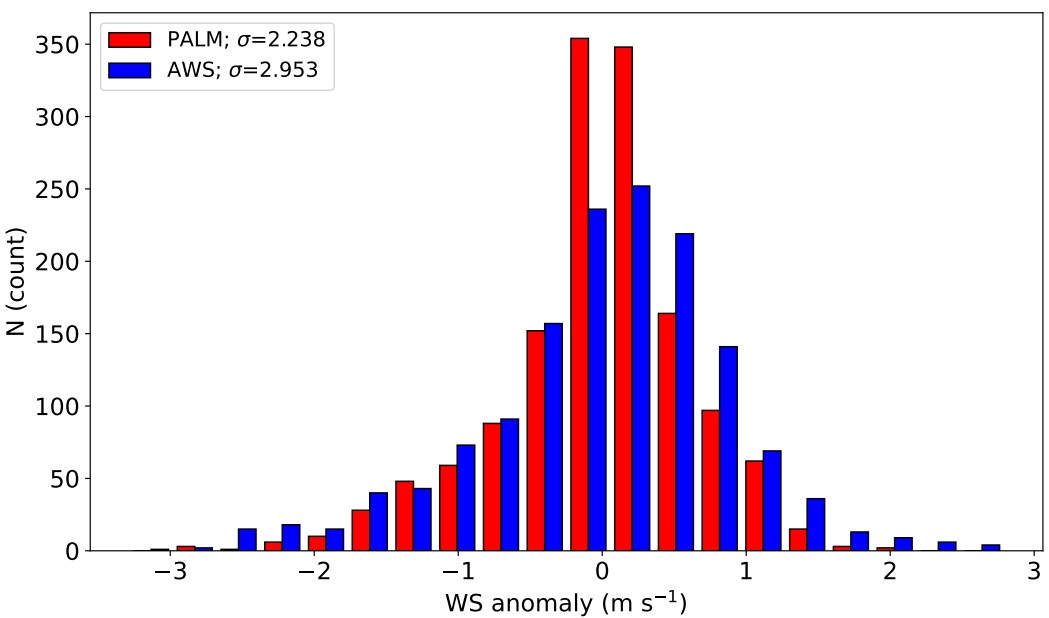

**Figure 14.** As in Figure 14, but for the north-easterly case.



**Table 1.** Variables in the PALM dynamic driver based on PALM input data standard v1.9.

| Variable Name | description | | |
|---|---|---|---|
| `init_soil_t` | Initial vertical profile of soil temperature | | |
| `init_soil_m` | Initial vertical profile of soil moisture | | |
| `init_atmosphere_X` | Initial vertical profile of X in the atmosphere as follows: | | |
| | **X** | **Variable long name** | **units** |
| | `pt` | Potential temperature | K |
| | `qv` | Specific humidity | $\text{kg kg}^{-1}$ |
| | `u` | Wind component in x-direction | $\text{m s}^{-1}$ |
| | `v` | Wind component in y-direction | $\text{m s}^{-1}$ |
| | `w` | Subsidence velocity | $\text{m s}^{-1}$ |
| `ls_forcing_left_X` | Large-scale forcing data of X for left model boundary | | |
| `ls_forcing_right_X` | Large-scale forcing data of X for right model boundary | | |
| `ls_forcing_north_X` | Large-scale forcing data of X for north model boundary | | |
| `ls_forcing_south_X` | Large-scale forcing data of X for south model boundary | | |
| `ls_forcing_ug` | $u$ wind component geostrophic (units: $\text{m s}^{-1}$) | | |
| `ls_forcing_vg` | $v$ wind component geostrophic (units: $\text{m s}^{-1}$) | | |
| `surface_forcing_surface_pressure` | Large-scale surface forcing of surface pressure (units: Pa) | | |





**Table 2.** Variables used in WRF4PALM.

| Variables | Units |
|---|---|
| Velocity components ($u$, $v$, $w$) | m s$^{-1}$ |
| Temperature | K |
| Potential temperature | K |
| Pressure | Pa |
| Water vapour mixing ratio | kg kg$^{-1}$ |
| Soil moisture | m$^3$ m$^{-3}$ |
| Soil temperature | K |
| Perturbation geopotential | m$^2$ m$^{-2}$ |
| Base-state geopotential | m$^2$ m$^{-2}$ |
| Latitudes and longitudes | degree |





**Table 3.** PALM domain configuration for the case studies.

| Directions | Total lengths | Grid points | Grid resolutions |
|:---:|:---:|:---:|:---:|
| X | 3600 m | 360 | 10 m |
| Y | 3600 m | 360 | 10 m |
| $Z^a$ | 2950 m | 200 | 10 m |

a: vertical grid resolution is trenched with a factor of 1.02 above z=1200 m with max

vertical grid resolution of 30 m





**Table 4.** Comparison of RMSE and IOA between the AWS observational data, the WRF modelling data, and the PALM modelling data at surface. Here wind indicates wind speed.

| Counterparts | Temperature RMSE | Temperature IOA | Wind RMSE | Wind IOA |
|---|---|---|---|---|
| **North-westerly Case** | | | | |
| AWS and WRF | 2.02 | 0.72 | 2.70 | 0.50 |
| AWS and PALM | 1.44 | 0.81 | 2.42 | 0.56 |
| WRF and PALM | 0.91 | 0.87 | 1.54 | 0.75 |
| **North-easterly Case** | | | | |
| AWS and WRF | 1.15 | 0.85 | 1.55 | 0.76 |
| AWS and PALM | 2.64 | 0.63 | 2.12 | 0.66 |
| WRF and PALM | 2.43 | 0.66 | 1.10 | 0.79 |
| **North-easterly Case before 0400 NZDT 16 February 2017** | | | | |
| AWS and WRF | 1.13 | 0.79 | 1.60 | 0.63 |
| AWS and PALM | 0.99 | 0.74 | 1.76 | 0.56 |
| WRF and PALM | 1.17 | 0.74 | 1.00 | 0.77 |
| **North-easterly Case after 0400 NZDT 16 February 2017** | | | | |
| AWS and WRF | 1.18 | 0.88 | 1.55 | 0.84 |
| AWS and PALM | 3.61 | 0.56 | 2.45 | 0.69 |
| WRF and PALM | 3.24 | 0.59 | 1.22 | 0.79 |