# Peer review of "WRF4PALM v1.0: A Mesoscale Dynamical Driver for the Microscale PALM Model System 6.0"

_Geoscientific Model Development, 2020_

## Referee Comment (RC1) · Anonymous Referee #1 · 14 Jan 2021

This manuscript is well-written and demonstrates the application of a dynamical driver for WRF output data to the PALM LES model. I recommend that the manuscript be Accepted after the authors address the following minor revisions:

Line 46: "other mesoscale models" are mentioned without a clear antecedent for "mesoscale models". I would recommend that the authors delete "other". If the authors choose to retain the word, it should be made clear whether PALM is a being referred to as a mesoscale model (which I would say is somewhat inaccurate and misleading), or the authors should indicate here or elsewhere what mesoscale model was previously discussed or referenced.

Lines 47-49: The sentence about "offline nesting" is difficult to follow. In most WRF model studies, this is often referred to as "one-way nesting", which I find to be some-

what clearer and more standard terminology. In any case, "...is realised as that mesoscale data are passed..." is an awkward construction that should be revised for clarity. If the authors choose to modify "offline-nesting", additional instances should also be addressed later in the manuscript.

Line 70-72: "meteorological forcing" is not completely consistent with "dynamical fields" (e.g. initial atmospheric profiles from the WRF are not dynamical, they are static inputs), and I think it should be aligned. I would recommend making "dynamical fields" more general and referring instead to "meteorological and sub-surface fields", "meteorological and soil outputs", or "meteorological and soil data" extracted from the WRF. If the authors choose to modify this terminology, other instances of "dynamical fields" should be addressed later in the manuscript as well.

Lines 155-156: This is related to the comment in Line 70 above, but referring to all possible WRF output as "mesoscale dynamics" feels too narrow to me. The point of this sentence is to indicate that only meteorological features that are resolved by the WRF can be passed to PALM, and parameterized processes are thus excluded from WRF4PALM. I would recommend that this sentence be revised to more effectively express this concept.

Line 160: There are minor grammatical errors in this section that should be addressed before final publication. They do not impede understanding, but they are mildly distracting.

Line 191: "resolutions" should be "grid spacings"

Line 204: "resolution" is unnecessary, or could be "spacing"

Line 206: It is not clear to me how the STG "reads synoptic conditions from the dynamic driver" when the dynamic driver is ingesting 1 km grid spacing WRF output data. Unless the WRF data is aggregated in time and space to yield "synoptic" conditions, I do not think this is accurate. The authors should clarify this point.

Line 237: It should be indicated/considered here and elsewhere that hourly average PALM and AWS data are being compared with instantaneous output from the WRF. I agree with the authors that variations in error/skill are likely attributable to cloud effects, but the potential for differences between hourly averages and instantaneous data to affect the statistical results should also be noted.

Fig. 2, caption: "resolution" should be "grid spacing"

Figs. 4 and 10: In these and other figures with an AWS label, it should be indicated in the captions that AWS refers to the observational data. Also, the temporal spacing for the wind direction data is different than the wind and temperature data. This difference should be noted in the caption, and in the discussions around Line 220 and (perhaps) around Line 297.

Fig. 14: The figure caption should read "As in Figure 8..."

[Figure]

---

## Referee Comment (RC2) · Anonymous Referee #2 · 7 Feb 2021

This article presents a set of Python tools for offline nesting from WRF into the PALM large-eddy simulation model (WRF4PALM). The authors describe the Python routines necessary to provide initial and lateral boundary conditions for PALM, and show two case studies for the urban environment of Christchurch in New Zealand, providing comparisons of 5-m wind speed and temperature to an automatic weather station. However, there is a number of fundamental reasons why I cannot recommend this work for publication in Geoscientific Model Development.

1) The authors make a case for not using the recently developed INFOR infrastructure by Kadasch et al. GMDD2020 (https://gmd.copernicus.org/preprints/gmd-2020-285/) based on its current applicability restricted to the COSMO, which is not open source model. Given that PALM is a community model, I question the decision of promoting a

duplicative tool instead of enhancing INFOR to accommodate WRF model data, which should in turn be more straightforward and at the same time more beneficial to the user community of PALM. This work appears to be redundant in that regard.

2) Within the WRF4PALM tools, the authors derive geostrophic wind components from WRF output to be used as forcing for PALM. This choice is not correct, as the lateral boundary conditions already provide the large-scale pressure gradient information implicitly, so this would be somewhat double counting of the mesoscale pressure gradient forcing. In addition, what about the ageostrophic component then? The current approach assumes that component is not relevant when that is often not the case. The use of geostrophic forcing in this context is not necessary and would lead to spurious mesoscale forcing driving PALM.

3) The authors mention the use of a synthetic turbulence generator (STG) to accelerate formation of resolved turbulence features. While this is a key component in the mesoscale-to-LES downscaling, the authors do not mention the specific method being used or if particular extensions have been made to accommodate realistic atmospheric flows. I suspect the authors are using Xie & Castro (2008) method, which is part of the PALM release. However, that approach was extended by Kadasch et al. GMDD2020 to include atmospheric stability information, as well as presented a comprehensive analysis of scaling and computational cost. This is again another strong argument why it does not seem a good idea to have WRF4PALM as a separate tool, rather the authors should leverage the INFOR preprocessor as already mentioned.

---

## Author Comment (AC1) · 1 Mar 2021

**Author's response for gmd-2020-306:**

**WRF4PALM v1.0: A Mesoscale Dynamical Driver for the Microscale PALM Model System 6.0**

The authors thank the reviewers for their time and consideration given to this manuscript. The reviewers' comments have been listed below in **bold** and responded to individually in *blue italics*. Revised sentences are in *red italics*. The marked-up manuscript is attached at the end.

**Reply to Referee #1**

**Line 46: "other mesoscale models" are mentioned without a clear antecedent for "mesoscale models". I would recommend that the authors delete "other". If the authors choose to retain the word, it should be made clear whether PALM is a being referred to as a mesoscale model (which I would say is somewhat inaccurate and misleading), or the authors should indicate here or elsewhere what mesoscale model was previously discussed or referenced.**

*We agree with this comment that the reference of "other mesoscale models" could be misleading. We have deleted the word "other" in Line 46.*

**Lines 47-49: The sentence about "offline nesting" is difficult to follow. In most WRF model studies, this is often referred to as "one-way nesting", which I find to be somewhat clearer and more standard terminology. In any case, "...is realised as that mesoscale data are passed. . ." is an awkward construction that should be revised for clarity. If the authors choose to modify "offline-nesting", additional instances should also be addressed later in the manuscript.**

*We agree with this comment. We have changed the sentences to:*

*"PALM was designed to seamlessly apply forcing from mesoscale models in a one-way or offline nesting approach (Maronga et al., 2015; Vollmer et al., 2015; Heinze et al., 2017; Maronga et al., 2020; Kadasch et al., 2020). Here one-way or offline nesting is realised as that meteorological forcing from mesoscale models are passed onto PALM, while PALM does not have to run along with or provide any feedback to the mesoscale model."*

**Line 70-72: "meteorological forcing" is not completely consistent with "dynamical fields" (e.g. initial atmospheric profiles from the WRF are not dynamical, they are static inputs), and I think it should be aligned. I would recommend making "dynamical fields" more general and referring instead to "meteorological and sub-surface fields", "meteorological and soil outputs", or "meteorological and soil data" extracted from the WRF. If the authors choose to modify this terminology, other instances of "dynamical fields" should be addressed later in the manuscript as well.**

*We agree with this comment and have changed "dynamical fields" to "meteorological and sub-surface fields".*

**Lines 155-156: This is related to the comment in Line 70 above, but referring to all possible WRF output as "mesoscale dynamics" feels too narrow to me. The point of this sentence is to indicate that only meteorological features that are resolved by the WRF can be passed to PALM, and parameterized processes are thus excluded from WRF4PALM. I would recommend that this sentence be revised to more effectively express this concept.**

*We followed the comment and have changed this sentence to:*

*The dynamic driver of PALM generated by the create_dynamic.py script only contains meteorological and sub-surface fields from WRF and does not encompass parameterised processes. Since turbulence is completely parameterised in WRF, it is also not included in the dynamic driver.*

**Line 160: There are minor grammatical errors in this section that should be addressed before final publication. They do not impede understanding, but they are mildly distracting.**

*We have revised Section 3.2.*

**Line 191: "resolutions" should be "grid spacings"**

*We followed the comment and have changed "resolutions" to "grid spacings"*

**Line 204: "resolution" is unnecessary, or could be "spacing"**

*We followed the comment and have changed "resolution" to "spacing"*

**Line 206: It is not clear to me how the STG "reads synoptic conditions from the dynamic driver" when the dynamic driver is ingesting 1 km grid spacing WRF output data. Unless the WRF data is aggregated in time and space to yield "synoptic" conditions, I do not think this is accurate. The authors should clarify this point.**

*In order to clarify how STG and dynamic driver work in PALM, we have added the following sentences in Line 206-213:*

*"The time-dependent 10 m grid spacing boundary conditions are stored in the dynamic driver processed by WRF4PALM."*

*"STG imposes perturbations on the boundary conditions given by the dynamic driver. STG embedded in PALM adopted the technique described by Xie and Castro (2008), which is described in detail in Kadasch et al. (2020)."*

**Fig. 2, caption: "resolution" should be "grid spacing"**

*"resolution" has been changed to "grid spacing"*

*Table 3 caption, "resolutions" have been changed to "spacings"*

**Figs. 4 and 10: In these and other figures with an AWS label, it should be indicated in the captions that AWS refers to the observational data. Also, the temporal spacing for the wind direction data is different than the wind and temperature data. This difference should be noted in the caption, and in the discussions around Line 220 and (perhaps) around Line 297.**

*We agree with this comment. In the captions of Figure 4 and Figure 8, we have added "(labelled as AWS)" where observational data are mentioned. In Figure 4 and 10, only 30-minute wind direction arrows are plotted in order to show the wind direction more clearly. If all 1-minute wind direction data are plotted as arrows, it would be hard to distinguish one dataset from another.*

*In Line 223, We have added*

*"In order to show wind direction clearly and avoid overlapping between data, only 30-minute wind direction PALM and observational data are shown in Figure 4."*

*In the caption of Figure 4, we have added:*

*"Only 30-minute PALM and observational data for wind direction are shown in (a) to avoid overlapping of arrows."*

**Fig. 14: The figure caption should read "As in Figure 8..."**

*This has been corrected.*

**Reply to Referee #2**

**The authors make a case for not using the recently developed INFOR infrastructure by Kadasch et al. GMDD2020 (https://gmd.copernicus.org/preprints/gmd-2020-285/) based on its current applicability restricted to the COSMO, which is not open source model. Given that PALM is a community model, I question the decision of promoting a duplicative tool instead of enhancing INFOR to accommodate WRF model data, which should in turn be more straightforward and at the same time more beneficial to the user community of PALM. This work appears to be redundant in that regard.**

*We thank the referee for bringing the INIFOR interface described by Kadasch et al. (GMDD 2020) to our attention. However, we have a few points to explain as why WRF4PALM makes a valuable contribution to the community:*

1. *Kadasch et al. (GMDD 2020) was not published when we submitted the manuscript. Hence, there was no published work describing any PALM mesoscale interface in detail at all. We now have added a citation of Kadasch et al. (GMDD 2020) in the manuscript:*
   a. *In Line 47 where one-way nesting between PALM and mesoscale models is mentioned*
   b. *In Line 50 where INIFOR is mentioned*
   c. *In Line 135 where the geostrophic wind forcing is mentioned*
   d. *In Line 212 where STG technique in PALM is mentioned*

2. *Before starting to work on the new interface, the WRF4PALM development team in New Zealand and Germany studied various options including using INIFOR for the WRF-PALM interface. After a careful assessment of this approach, we decided to develop a standalone interface independent of INIFOR and its static driver, for the following reasons:*
   a. *INIFOR by its structure and design has many limitations. INIFOR is mostly a hard-coded utility, which is difficult to be amended, changed or extended. Additionally, INIFOR has a fixed domain that is setup mostly over central Europe. Therefore, INIFOR is not a straightforward tool that can directly process data sets from other models especially when regions outside Europe (e.g. New Zealand as in this work) are studied.*
   b. *INIFOR is written in Fortran which is fairly difficult to read and modify. In contrast, Python is a free, open source and modern language offering many features. Python is hence the choice of language for a lot of scientific code development. Python is a cross-functional, maximally interpreted language. Due to its high code readability, easy syntax and object-oriented programming approach, Python code is easy to understand and modify. A huge Python community, user support and extensive package library makes python language of choice for programmers. Hence, we chose to develop WRF4PALM using Python rather than extending INIFOR. Since the discussion paper of WRF4PALM was made available online, we have already received a request from a research group in Germany wanting to use WRF4PALM and contribute to its development. This already shows the significance of having WRF4PALM developed and easily accessible.*
   c. *The static driver which provides geospatial surface data to the PALM model is essential for developing the dynamic driver. The static driver code provided by the PALM team (palm_csd and palm_csd_files, Heldens et al. 2020) is already written in Python. Therefore, with the implementation of WRF4PALM, both static driver and dynamic driver are using a consistent programming language, which saves the learning hurdle for PALM users.*

3. *PALM and WRF are both community models. WRF4PALM, therefore, provides a better chance for both communities to expand their research applications. This is especially true with WRF4PALM's open source and widely used Python implementation. Developing multiple tools in parallel is very common in the scientific world. More than one tools can be developed using different approaches, programming languages, and algorithms. This provides greater flexibility, a wider range of solutions, convenience, and ease of use to the users.*

4. *Although Kadasch et al. (GMDD 2020) mentioned to extend INIFOR to process WRF and ICON output, it is unclear when the implementation will be done and whether INIFOR will be able to process*

*geographical domains outside Europe. In addition, with the current implementation of WRF4PALM, it is quite easy and straightforward to include chemistry fields from WRF-Chem and this is the next step of our future WRF4PALM development. It is unclear whether such a development is already planned for INIFOR.*

**Within the WRF4PALM tools, the authors derive geostrophic wind components from WRF output to be used as forcing for PALM. This choice is not correct, as the lateral boundary conditions already provide the large-scale pressure gradient information implicitly, so this would be somewhat double counting of the mesoscale pressure gradient forcing. In addition, what about the ageostrophic component then? The current approach assumes that component is not relevant when that is often not the case. The use of geostrophic forcing in this context is not necessary and would lead to spurious mesoscale forcing driving PALM.**

*We adapted the method regarding the geostrophic winds from INIFOR as described on PALM's official website before the preprint of Kadasch et al. (GMDD 2020) was available online. The fact that velocity fields can be separated into geostrophic and ageostrophic components does not imply that ageostrophic components are missing in WRF4PALM. The geostrophic winds are not used to reconstruct velocity fields. Rather, the geostrophic wind definition is only used to represent the large-scale pressure gradient in a convenient way in terms of velocity components. A similar approach is also used by INIFOR (page 16 in Kadasch et al. (GMDD 2020)).*

*Geostrophic forcing is currently not required for the mesoscale interface. Since WRF4PALM is completely open-source, the users of WRF4PALM can make their own decision on whether to include the geostrophic wind or not by amending the source code themselves. We will add this feature in future development of WRF4PALM.*

*We now further direct our readers to Kadasch et al. (2020) where a short discussion is presented. This sentence has been added in Line 135 for the geostrophic wind forcing:*

*Note that as commented in the PALM 6.0 Overview (Maronga et al., 2020) and in the PALM mesoscale nesting interface description (Kadasch et al., 2020), geostrophic winds are not required in the dynamic driver at present. Users can exclude the geostrophic wind forcing by amending the WRF4PALM code themselves.*

**The authors mention the use of a synthetic turbulence generator (STG) to accelerate formation of resolved turbulence features. While this is a key component in the mesoscale-to-LES downscaling, the authors do not mention the specific method being used or if particular extensions have been made to accommodate realistic atmospheric flows. I suspect the authors are using Xie & Castro (2008) method, which is part of the PALM release. However, that approach was extended by Kadasch et al. GMDD2020 to include atmospheric stability information, as well as presented a comprehensive analysis of scaling and computational cost. This is again another strong argument why it does not seem a good idea to have WRF4PALM as a separate tool, rather the authors should leverage the INFOR preprocessor as already mentioned.**

*The Synthetic Turbulence Generator (STG) is an independent part of the PALM model, it is not developed, modified or re-implemented as part of INIFOR or WRF4PALM. We used STG in the same way that it is used with INIFOR. Kadasch et al. (GMDD, 2020) also employed Xie and Castro (2008) method (see their Section 2.3). The scaling and computational cost would also be up to the correspondent PALM STG development team instead of WRF4PALM. Our work only focuses on mesoscale-to-LES downscaling from WRF to PALM rather than the STG.*

[revised manuscript text omitted]

+: vertical grid spacing is trenched with a factor of 1.02 above z=1200 m with max vertical grid spacing of 30 m

**Table 4.** Comparison of RMSE and IOA between the AWS observational data, the WRF modelling data, and the PALM modelling data at surface. Here wind indicates wind speed.

| Counterparts | Temperature RMSE | Temperature IOA | Wind RMSE | Wind IOA |
|---|---|---|---|---|
| **North-westerly Case** | | | | |
| **AWS and WRF** | 2.02 | 0.72 | 2.70 | 0.50 |
| **AWS and PALM** | 1.44 | 0.81 | 2.42 | 0.56 |
| **WRF and PALM** | 0.91 | 0.87 | 1.54 | 0.75 |
| **North-easterly Case** | | | | |
| **AWS and WRF** | 1.15 | 0.85 | 1.55 | 0.76 |
| **AWS and PALM** | 2.64 | 0.63 | 2.12 | 0.66 |
| **WRF and PALM** | 2.43 | 0.66 | 1.10 | 0.79 |
| **North-easterly Case before 0400 NZDT 16 February 2017** | | | | |
| **AWS and WRF** | 1.13 | 0.79 | 1.60 | 0.63 |
| **AWS and PALM** | 0.99 | 0.74 | 1.76 | 0.56 |
| **WRF and PALM** | 1.17 | 0.74 | 1.00 | 0.77 |
| **North-easterly Case after 0400 NZDT 16 February 2017** | | | | |
| **AWS and WRF** | 1.18 | 0.88 | 1.55 | 0.84 |
| **AWS and PALM** | 3.61 | 0.56 | 2.45 | 0.69 |
| **WRF and PALM** | 3.24 | 0.59 | 1.22 | 0.79 |